# A rotifer-derived paralytic compound prevents transmission of schistosomiasis to a mammalian host

Jiarong Gao[1,2☯], Ning Yang[3☯], Fred A. Lewis[4¤a], Peter Yau[5], James J. Collins, III[2¤b], Jonathan V. Sweedler[3]*, Phillip A. Newmark[2,6,7,8]*

1 Cellular and Molecular Biology Program, University of Wisconsin-Madison, Madison, Wisconsin, United States of America, 2 Department of Cell and Developmental Biology, University of Illinois at Urbana-Champaign, Urbana, Illinois, United States of America, 3 Department of Chemistry and the Beckman Institute, University of Illinois at Urbana-Champaign, Urbana, Illinois, United States of America, 4 Biomedical Research Institute, Rockville, Maryland, United States of America, 5 Roy J. Carver Biotechnology Center, University of Illinois at Urbana-Champaign, Urbana, Illinois, United States of America, 6 Howard Hughes Medical Institute, Chevy Chase, Maryland, United States of America, 7 Morgridge Institute for Research, Madison, Wisconsin, United States of America, 8 Department of Integrative Biology, University of Wisconsin-Madison, Madison, Wisconsin, United States of America

☯ These authors contributed equally to this work.
¤a Current address: Bernville, Pennsylvania, United States of America
¤b Current address: Department of Pharmacology, UT Southwestern Medical Center, Dallas, Texas, United States of America
* jsweedle@illinois.edu (JVS); pnewmark@morgridge.org (PAN)

**Data Availability Statement:** The data files for NMR, LC-MS, MALDI MS and H/D exchange MS have been deposited into Illinois DataBank. DOI: https://doi.org/10.13012/B2IDB-1599850_V1.

## Abstract

Schistosomes are parasitic flatworms that infect over 200 million people, causing the neglected tropical disease, schistosomiasis. A single drug, praziquantel, is used to treat schistosome infection. Limitations in mass drug administration programs and the emergence of schistosomiasis in nontropical areas indicate the need for new strategies to prevent infection. It has been known for several decades that rotifers colonizing the schistosome's snail intermediate host produce a water-soluble factor that paralyzes cercariae, the life cycle stage infecting humans. In spite of its potential for preventing infection, the nature of this factor has remained obscure. Here, we report the purification and chemical characterization of Schistosome Paralysis Factor (SPF), a novel tetracyclic alkaloid produced by the rotifer *Rotaria rotatoria*. We show that this compound paralyzes schistosome cercariae and prevents infection and does so more effectively than analogous compounds. This molecule provides new directions for understanding cercariae motility and new strategies for preventing schistosome infection.

## Introduction

Schistosomiasis—caused by parasitic flatworms of the genus *Schistosoma*—is a major neglected tropical disease, affecting over 200 million people, with over 700 million people at risk of infection [1–3]. Praziquantel is currently the only drug used for treating schistosomiasis. Concerns

**Funding:** This work was supported by: Howard Hughes Medical Institute (https://www.hhmi.org/): Investigator Award to PAN; International Student Research Fellowship to JG; National Institute of Neurological Diseases and Stroke (https://www.ninds.nih.gov/): R01 NS031609 to JVS; National Institute on Drug Abuse (https://www.drugabuse.gov/): P30 DA018310 to JVS. The funders had no role in study design, data collection and analysis, decision to publish, or preparation of the manuscript.

**Abbreviations:** ACN, acetonitrile; APW, artificial pond water; BRI, Biomedical Research Institute; CHCA, alpha-cyano-4-hydroxy-cinnamic acid; COSY, correlation spectroscopy; FA, formic acid; FTMS, Fourier-transform mass spectrometry; gCOSY, gradient selected correlation spectroscopy; GPCR, G protein-coupled receptor; HMBC, heteronuclear multiple-bond correlation; HPLC, high-performance liquid chromatography; HSQC, heteronuclear single quantum coherence spectroscopy; M+H, protonated molecular ion; MALDI-MS, matrix-assisted laser desorption/ionization mass spectrometry; MS, mass spectrometry; MWCO, molecular weight cut-off; NMR, nuclear magnetic resonance; NOE, nuclear Overhauser effect; NOESY, nuclear Overhauser effect spectroscopy; PBS, phosphate-buffered saline; Q-TOF, quadrupole time-of-flight; RP-HPLC, reversed-phase high-performance liquid chromatography; SPF, Schistosome Paralysis Facto; TOCSY, total correlation spectrometry.

about the emergence of drug resistance [4,5], as well as limitations observed in mass drug administration programs [6–9], highlight the need to devise new strategies for preventing infection by these parasites. This need is amplified by the recent identification of people infected with human-livestock hybrid schistosomes and the geographical expansion of schistosomiasis to temperate regions [10–12].

Schistosomes have a complex life cycle that alternates between an intermediate host (snail) and a definitive host (mammal) via 2 free-living, water-borne forms called miracidia and cercariae, respectively [13] (Fig 1A). For decades, inconsistency in cercarial production by snails and infectivity of mammalian hosts has been observed in most schistosome laboratories [14]. Intriguingly, Stirewalt and Lewis reported that rotifer colonization on shells of the snail intermediate host (*Biomphalaria glabrata*) significantly reduced cercariae output, motility, and infectivity [15]. Furthermore, they observed that cercarial motility was affected not only by the presence of rotifers but also by rotifer-conditioned water, indicating that rotifers released water-soluble molecules with paralytic activity. Almost 40 years have passed since this important finding, yet this factor's identity has remained a mystery.

## Results and discussion

### Purification of the rotifer-derived compound

Encouraged by this anticercarial effect and its potential to prevent schistosome infection, we sought to purify this paralyzing agent. We isolated individual rotifers from snail shells and found 2 species, *R. rotatoria* (Fig 1B) and *Philodina acuticornis* (Fig 1C), as previously reported by Stirewalt and Lewis [15]. To identify which rotifer was responsible for the paralytic effect, we grew clonal isolates of each species, producing rotifer-conditioned artificial pond water (APW). Adding *R. rotatoria*-conditioned APW to freshly collected cercariae resulted in gradual paralysis within 5 min (Fig 1D). Most cercariae stopped swimming and sank to the bottom of the dish. Tapping the dish could stimulate their movement, but their response was limited to writhing on the dish bottom or short-distance swimming before becoming paralyzed again. In contrast, *P. acuticornis*–conditioned water had no effect (Fig 1E).

To purify the paralyzing agent, we performed molecular weight cut-off (MWCO) filtration of rotifer-conditioned water and found that the activity was present in the <650 Da fraction. The <650 Da filtrate was fractionated by reversed-phase high-performance liquid chromatography (RP-HPLC; Fig 2A), and each fraction was tested on cercariae. Paralysis was only observed following treatment with a peak eluting at 25 to 27 min (Fig 2B). As expected, this peak was detected only in *R. rotatoria*–but not *P. acuticornis*–conditioned water (Fig 2B). A second round of HPLC on this peak revealed one peak (eluting at 24–26 min) with paralytic activity (Fig 2C). A predominant signal of 273.16 Da (protonated molecular ion [M+H]) in this peak was revealed by matrix-assisted laser desorption/ionization mass spectrometry (MALDI-MS; Fig 2D). Consistent with the paralysis assay, this signal (*m/z* 273.16) was detected exclusively in the fraction eluting at 24 to 26 min but not in the fractions before or after (Fig 2E). These results suggested that the component with *m/z* 273.16 was the paralyzing agent, which we named "Schistosome Paralysis Factor" (SPF). We then determined the monoisotopic mass for protonated SPF using high-resolution quadrupole time-of-flight (Q-TOF) MS, 273.1595 Da (Fig 2F), suggesting $C_{16}H_{20}N_2O_2$ as the best-fitting formula for SPF.

### SPF is a novel tetracyclic alkaloid

To elucidate its structure, we purified approximately 0.1 mg SPF from 25 L *R. rotatoria*–conditioned water. Nuclear magnetic resonance (NMR) spectroscopy revealed a novel tetracyclic structure. Briefly, [1]H spectra showed the presence of 19 protons in the compound (S1 Fig),

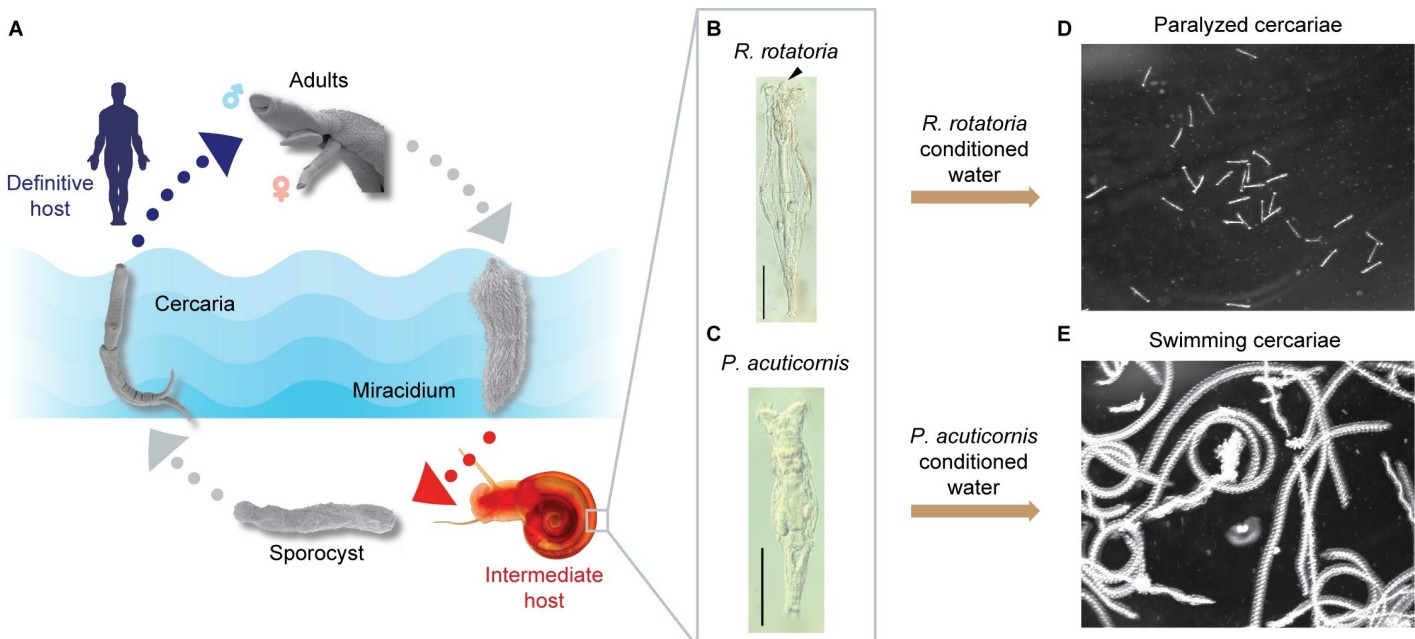

**Fig 1. *R. rotatoria*–conditioned water paralyzes *Schistosoma mansoni* cercariae.** (A) Life cycle of *S. mansoni*. Adult parasites, residing in the mammalian host vasculature, lay eggs (not shown). Upon exposure to fresh water, eggs release miracidia, which infect the appropriate snail host. Inside the snail, the parasite reproduces asexually, ultimately producing large numbers of free-swimming infective larvae (cercariae) that can penetrate mammalian skin to continue the life cycle (adapted from [16]). (B and C) Nomarski differential interference contrast microscopy images of *R. rotatoria* and *Philodina acuticornis* (arrowhead indicates the rostrum in *R. rotatoria*, which is lacking in *P. acuticornis*). Scale bars: 100 μm. (D and E) Maximum intensity projection (5 s, 150 frames) of cercariae motility after treatment with *R. rotatoria*– or *P. acuticornis*–conditioned water.

which agrees with the best-fitting formula and Hydrogen/Deuterium exchange mass spectrometry (MS) analysis (S2 Fig). Heteronuclear single quantum coherence spectroscopy (HSQC) revealed 3 methyl, 2 methylene, 6 methine groups, and 5 quaternary carbons (S3 Fig). Total correlation spectrometry (TOCSY) showed that aliphatic protons, except 2 methyl groups, are from one spin system (S4 Fig). The connectivity of the neighboring groups was derived from correlation spectroscopy (COSY) and heteronuclear multiple-bond correlation (HMBC) spectra (S5 Fig and S6 Fig). Overall, the aliphatic region is composed of a dimethyl-pyrrolidine structure, which is linked to an indole via a $CH_2$ group and an oxygen. Nuclear Overhauser effect spectroscopy (NOESY) suggested (*R*, *S*, *S*) or (*S*, *R*, *R*) configurations on the chiral centers (S7 Fig). Altogether, combined NMR analysis led to 2 possible structures (Fig 2G and 2H and S1 Table).

## SPF and its analogs paralyze cercariae in a dose-dependent manner

To test its dose dependency, we examined the paralytic effect of serially diluted SPF on cercariae by quantifying their movement over time. In the absence of SPF, over 82% of cercariae were free swimming over 3 min (Fig 3A and S1 Data). In 2.5 nM SPF, the percentage of free-swimming cercariae dropped to 67% 3 minutes after drug treatment. As the concentration of SPF increased, so did the rate of paralysis, and more cercariae were paralyzed at the end of treatment. We observed maximum effects in 250 nM and 2.5 μM SPF, with the majority of cercariae paralyzed within 30 s.

Two natural compounds from *Streptomyces* sp., ht-13-A and ht-13-B, are structurally related to SPF; they were isolated based on their affinities for human serotonin receptors [17]. All 3 alkaloids share a novel oxepineindole framework fused with a pyrrolidine ring (Fig 3A–

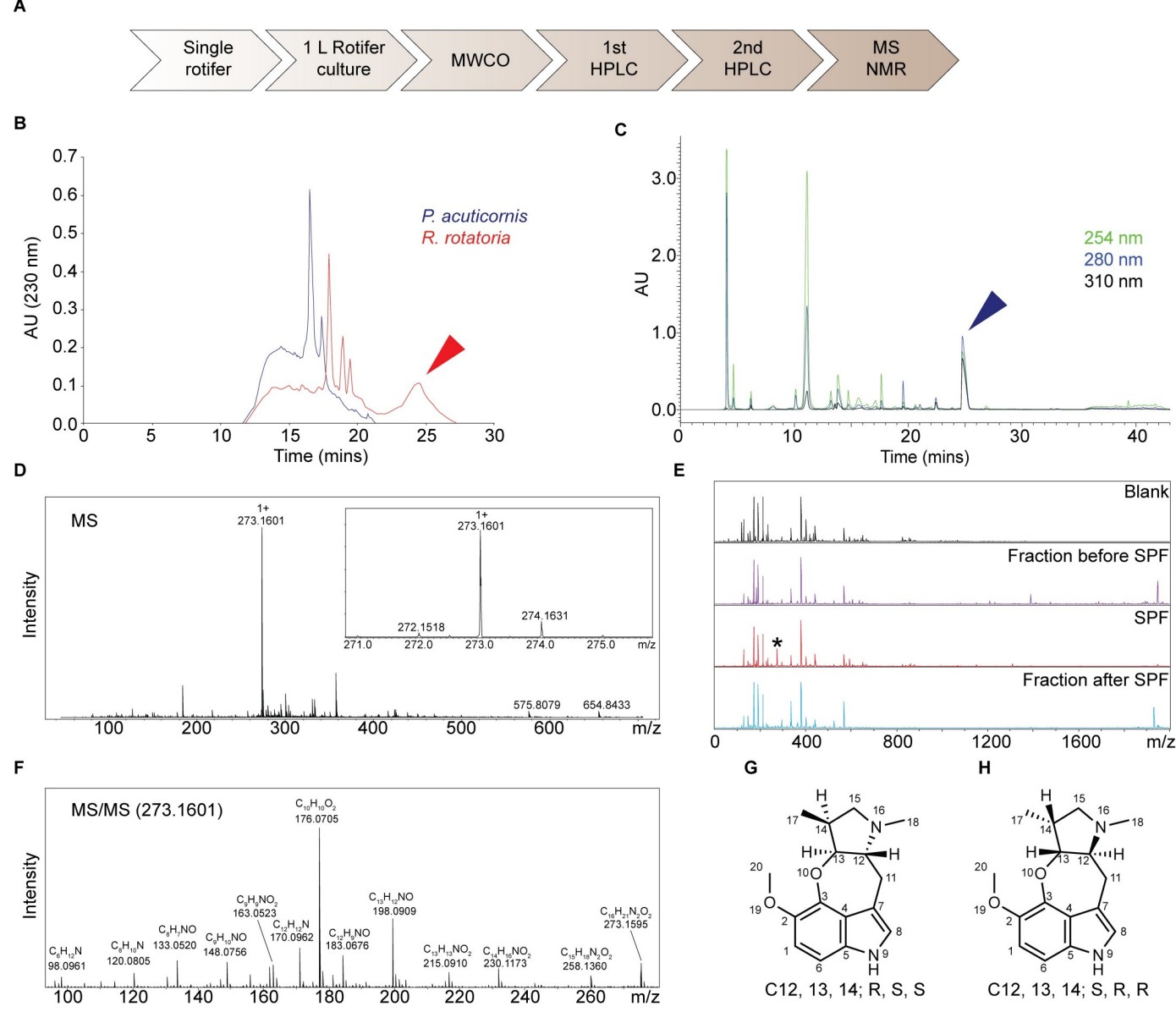

**Fig 2. SPF is a novel tetracyclic alkaloid.** (A) Flowchart for SPF purification. (B) First HPLC plots of *R. rotatoria*–and *P. acuticornis*–conditioned water. All fractions were tested for bioactivity; the red arrowhead indicates the only active peak. (C) Second HPLC plot of the bioactive fraction (red arrowhead in panel B). All peaks were tested for bioactivity; the blue arrowhead indicates the only peak containing activity. (D) MS showing the dominant signal of *m/z* 273.1601 from the peak (blue arrowhead). (E) MS plots showing this signal (asterisk, *m/z* 273.1601) was only detected in the fraction eluting at 24 to 26 min. (F) Tandem MS acquired from high-resolution Q-TOF analysis. (G and H) NOESY resolved the relative stereochemistry of 3 chiral centers and narrowed it down to 2 possible configurations. HPLC, high-performance liquid chromatography; MS, mass spectrometry; MWCO, molecular weight cut-off; NMR, nuclear magnetic resonance; NOESY, Nuclear Overhauser effect spectroscopy; Q-TOF, quadrupole time-of-flight; SPF, Schistosome Paralysis Factor.

3C and S1 Data; note the serotonin backbone highlighted in red in Fig 3A). Although synthesis of SPF has not been achieved, total syntheses of ht-13-A and ht-13-B have been reported [18–20]. To test whether this shared tetracyclic scaffold is responsible for the paralytic effect, we analyzed structure-activity relationships by using ht-13-A, ht-13-B, 3 ht-13-A derivatives [18], and one epimer in cercarial paralysis assays. Importantly, ht-13-A, although not as potent as SPF, also had a paralytic effect on cercariae (Fig 3B and S1 Data). In contrast, ht-13-B did not paralyze cercariae, suggesting that the extra methyl group disrupts interaction with the target

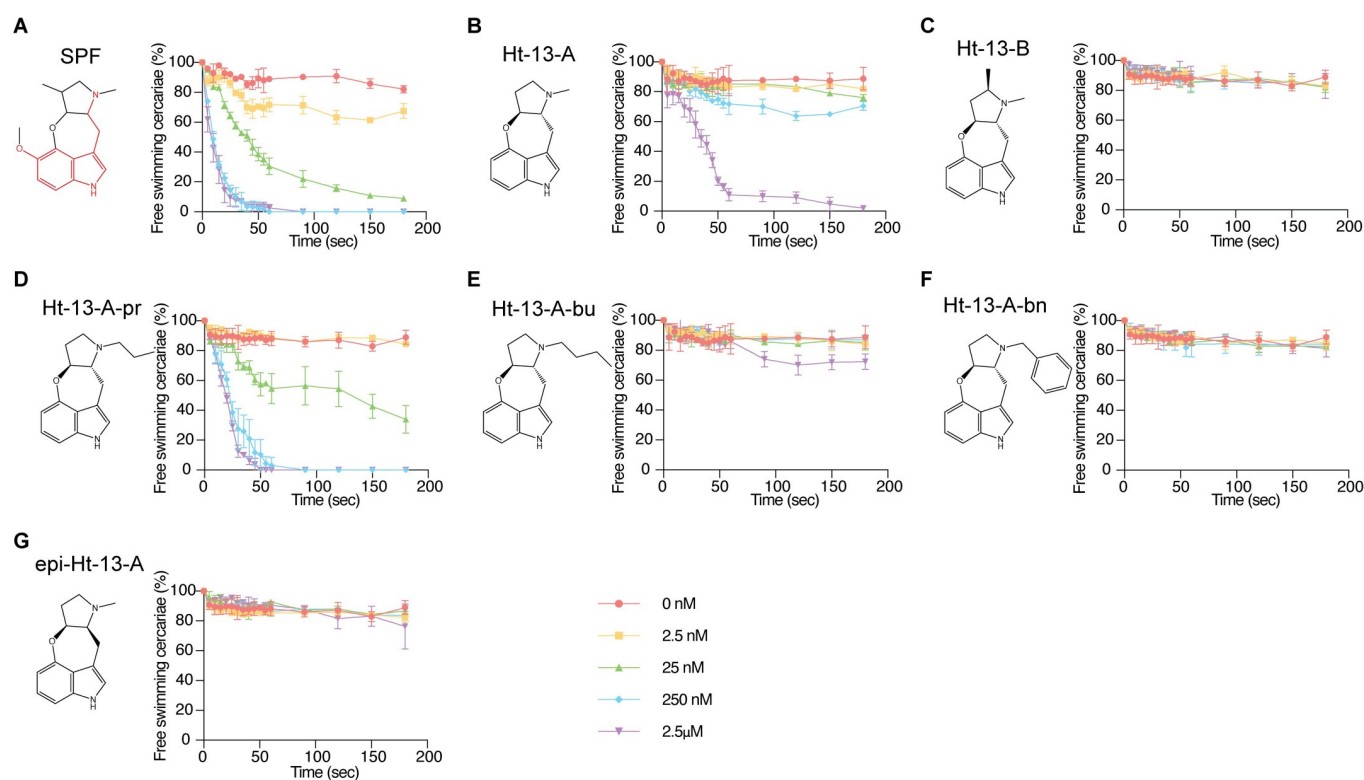

**Fig 3. Structure-activity relationships of SPF and related compounds as measured by cercarial motility assays.** (A–G) Percentage of cercariae (approximately 50) continuing to swim over 3 min after addition of each compound at specified final concentrations. Triplicates were performed. Data are mean ± SD. Serotonin structure in SPF is outlined in red. See S1 Data for corresponding raw data. SPF, Schistosome Paralysis Factor.

(Fig 3C and S1 Data). Of the 3 ht-13-A analogs, only ht-13-A-pr effectively paralyzed cercariae; it was more potent than ht-13-A, indicating that the nature of the side chain is important for proper target interaction (Fig 3D and 3E and S1 Data). In contrast to ht-13-A and Ht-13-A-pr, the epimer was unable to paralyze cercariae; these results support the (R, S) configuration of SPF at C12, 13 as the biologically active form (Fig 2G).

## SPF prevents mammalian infection

Because motility of the cercarial tail is essential for swimming and provides force for skin penetration [21–23], we examined whether SPF prevented infection. We treated approximately 200 cercariae with different concentrations of SPF for 10 min and then tested their infectivity by exposing them to mouse tails for 30 min (N = 6 for each condition). Six weeks post infection, we euthanized the mice, counted schistosomes recovered after hepatic portal vein perfusion, and examined liver pathology. From controls, we recovered 83 adult worms on average (Fig 4B and S2 Data), consistent with typical recoveries of approximately 40% [24]. Livers from these mice appeared dark and contained extensive granulomas (Fig 4A). In contrast, we did not recover any adult worms from mice after treatment with 250 nM or 2.5 μM SPF (Fig 4B and S2 Data), and no granulomas were observed (Fig 4A). Histological examination confirmed that these livers were free of schistosome eggs (Fig 4E and S2 Data), suggesting complete inhibition of infection. These data are consistent with the full paralysis observed after treatment with 250 nM or 2.5 μM SPF (Fig 3A and S1 Data). Although 25 nM SPF paralyzed most cercariae in vitro, the effects on mouse infection were not as severe (Fig 4A). Mechanical

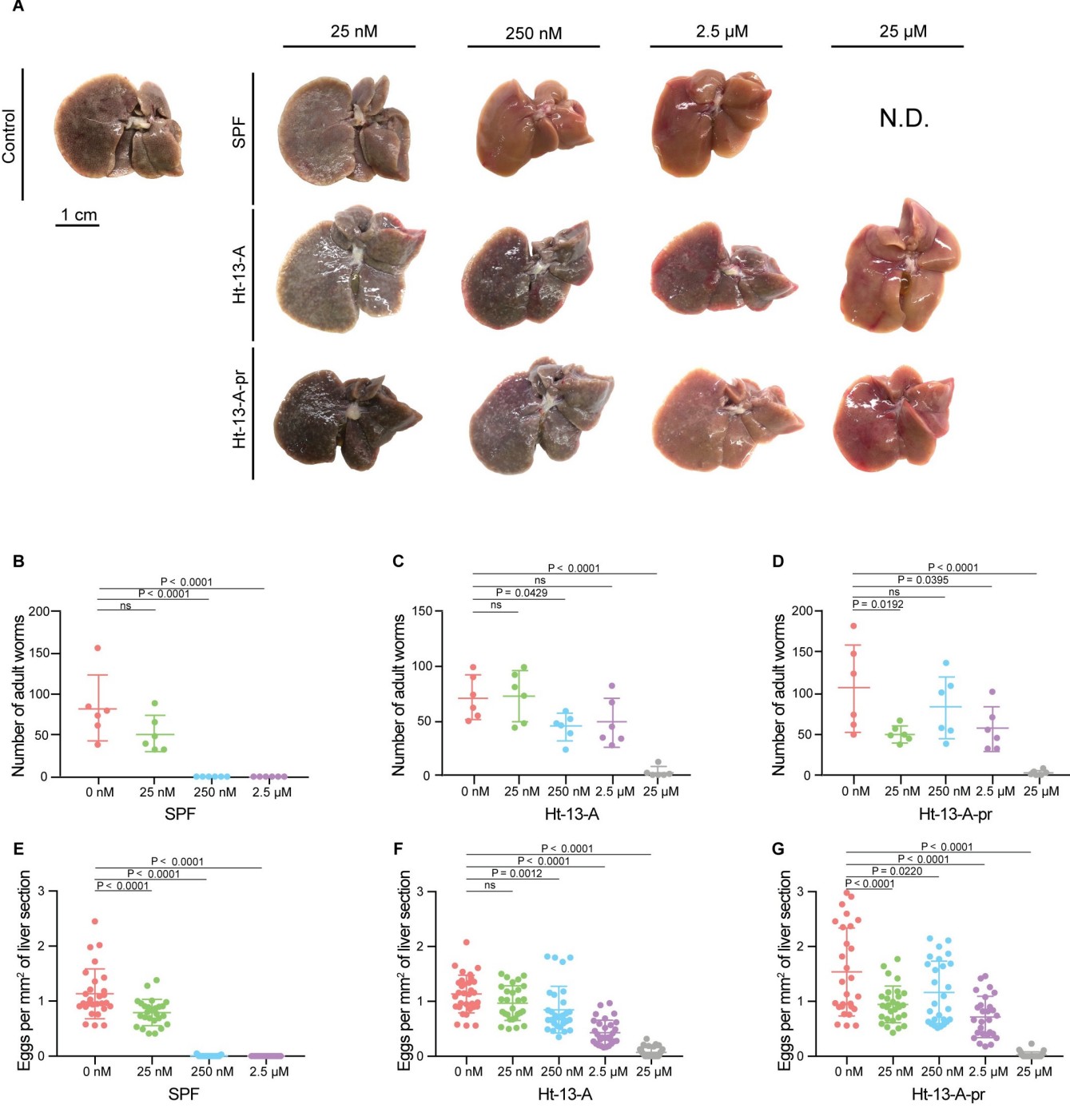

**Fig 4. Treating cercariae with SPF, Ht-13-A, or Ht-13-A-pr blocks schistosome infection and alleviates pathology.** (A) Representative livers (post perfusion) from mice ($N = 6$) exposed to drug-treated cercariae. Livers from mice treated with control and lower drug concentrations were darker in color and contained more granulomas (white spots). With higher drug concentrations, livers had normal morphologies with few or no granulomas; 25 μM SPF treatment was ND because of limited amounts of purified SPF. (B–D) Numbers of adult worms recovered from exposed mice (2 experiments for each drug, 6 mice total for each condition). (E–G) Numbers of schistosome eggs per area ($/mm^2$) from liver sections (4–6 sections per mouse). Data for panels B–G are mean ± SD. Statistics: One-way ANOVA, post Dunnett's test. See S2 Data for corresponding raw data. ND, not determined; SPF, Schistosome Paralysis Factor.

and/or chemical stimuli from mouse tails may overcome SPF-induced paralytic effects at low SPF concentrations. Notably, neither Ht-13-A nor Ht-13-A-pr blocked infection as completely

as 250 nM SPF, even at 25 μM (Fig 4A, 4C, 4D, 4F and 4G and S2 Data). Under more realistic infection conditions, in which mouse tails were lifted 1 to 2 cm from the bottom of the test tube containing cercariae, so they had to swim actively towards the tail to infect the mouse, Ht-13-A and Ht-13-A-pr were still not as effective as SPF, which completely blocked infection (S8 Fig).

## Conclusion

This work has identified a novel tetracyclic alkaloid, produced by the rotifer *R. rotatoria*, that paralyzes the infective larvae of schistosomes. Although its mechanism of action remains unknown, its chemical structure provides important clues. SPF contains a serotonin backbone, suggesting that SPF might antagonize serotonin signaling, perhaps via G protein-coupled receptors (GPCRs) or serotonin-gated channels. Consistent with this idea, the structurally related compounds, ht-13-A and ht-13-B, bind several human serotonin receptors [17]. In schistosomes, serotonin has been implicated in neuromuscular functions in multiple life cycle stages [25–28]; knocking down a serotonergic GPCR (Sm5HTR) in schistosomulae and adult worms led to decreased movement [29]. Interestingly, praziquantel partially activates the human serotonin receptor, HT2BR, suggesting that it may also target schistosome serotonergic GPCRs [30].

The chemical ecology underlying *R. rotatoria*'s production of SPF is also unclear. Whether SPF is used naturally to combat other aquatic creatures (e.g., to prevent other rotifers from colonizing areas where *R. rotaria* live) and, thus, the effect on schistosome cercariae is indirect, or whether SPF benefits the rotifer's commensal host will require further study. Because compounds with structural similarities to SPF are produced by *Streptomyces* sp., it will be important to examine the possibility that SPF is not directly produced by the rotifer but rather by constituent(s) of its own microbiome. However, given that horizontal gene transfer is well documented in rotifers [31,32], it is also possible that *R. rotatoria* has acquired the synthetic machinery to produce SPF on its own. Future work will help reveal the source of SPF and its biosynthetic pathway.

In the past few decades, the discovery and development of natural products have helped combat parasitic diseases [33]. Based on its ability to block infection, SPF holds great promise as an antischistosomal agent. Identifying the biologically active chemical scaffolds and understanding SPF's mode of action are expected to provide important clues for preventing schistosomiasis.

## Materials and methods

### APW

Four stock solutions were prepared to make APW [34]: (1) 0.25 g/L $FeCl_3 \cdot 6H_2O$, (2) 12.9 g/L $CaCl_2 \cdot 2H_2O$, (3) 10 g/L $MgSO_4 \cdot 7H_2O$, and (4) 34 g/L $KH_2PO_4$ 1.5 g/L $(NH_4)_2SO_4$ (pH 7.2) (Sigma-Aldrich; St. Louis, MO),. For 1 L APW, we added 0.5 mL of $FeCl_3$ solution, 2.5 mL $CaCl_2$ solution, 2.5 mL $MgSO_4$ solution, and 1.25 mL phosphate buffer.

### Obtaining *S. mansoni* cercariae

Infected *B. glabrata* snails provided by Biomedical Research Institute (BRI; Rockville, MD) were maintained in APW and fed Layer Crumbles (chicken feed; Rural King, Mattoon, IL). To obtain *S. mansoni* cercariae, *B. glabrata* snails were exposed to light at 26°C for 1 to 2 h. APW containing cercariae was passed through a 100 μm cell strainer (Falcon; Corning, NY) to

remove snail food and feces. Cercariae were then collected using custom-made 20 μm cell strainers.

## Rotifer culture

Because both rotifer species reproduce parthenogenetically, we clonally expanded each species into 1 L cultures from a single rotifer. Individual rotifers (*R. rotatoria* and *P. acuticornis*) were initially isolated from the shell of *B. glabrata* and cultured in APW in 24-well plates. Each individual colony was expanded into ever-larger culture volumes and ultimately maintained in 2 L flasks. Both species were fed Roti-rich liquid invertebrate food (Florida Aqua Farms Inc.; Dade City, FL). Rotifer-conditioned water was collected every month by filtering out the rotifers using a 20-μm cell strainer. Filtered rotifers were then passaged to fresh APW to propagate the cultures.

## Crude rotifer-conditioned water preparation

One-liter rotifer media was lyophilized, reconstituted with 50 mL $dH_2O$, and filtered through 10,000 and 650 MWCO Pall Minimate TFF Capsules with Omega membrane (Ann Arbor, MI). Filtrate (<650 Da) was freeze dried. For RP-HPLC, 300 mg of the dried material was dissolved in $dH_2O$ and run on a RP-HPLC—Merck Chromolith semi-prep RP-18e column (Darmstadt, Germany) at 5 ml/min using a gradient of 100% A (water) to 60% B (acetonitrile; ACN) in 60 min. A total of 10 mL fractions were collected and assayed for biological activity. Fractions containing biological activity were saved for further study.

## Further purification of rotifer media

The bioactive fractions were pooled, freeze dried with SpeedVac (Savant, MA), reconstituted with 500 μL $dH_2O$, and injected into a 4.6 mm diameter × 25 cm Symmetry column (Waters; Millford, MA). A Breeze2 analytical LC system (Waters; Millford, MA) was employed for separation at 0.5 mL/min with the following solvents and gradients: Solvent A, 0.1% formic acid (FA); solvent B, methanol with 0.1% FA; 0 to 10 min 0% to 10% B, 10 to 30 min 10% to 35% B, 30 to 33 min 35% to 80% B, 33 to 37 min 80% to 80% B, 37 to 40 min 80% to 0% B. Eluents were collected manually based on peak elution. All fractions were lyophilized, reconstituted with water, and analyzed with MALDI-MS. Fractions containing biological activity were saved for future use.

## MALDI-MS analysis

For each collected fraction, 1 μL of sample solution was spotted on a ground steel MALDI target and mixed with 1 μL of alpha-cyano-4-hydroxy-cinnamic acid (CHCA; Sigma-Aldrich; St. Louis, MO) solution (10 mg/mL CHCA in 50% ACN solution with 0.005% trifluoroacetic acid). Mass calibration, spectra acquisition, and analysis were performed under conditions as previously described by Tillmaand and colleagues [35].

## High-resolution Q-TOF MS analysis

A total of 1 μL of the bioactive fraction was separated on a Magic 0.1 × 150mm column (Michrom, CA) and analyzed with a maXis 4G mass spectrometer (Bruker; Billerica, MA) using previously established methods for metabolite study [36]. The separation was performed at 300 nl/min by use of solvent A (95% water, 5% ACN with 0.1% FA) and solvent B (5% water, 95% ACN with 0.1% FA) with the following gradient conditions: 0 to 5 min 4% B, 5 to

50 min 4% to 50% B, 50 to 52 min 50% to 90% B, 52 to 60 min 90% B, 60 to 70 min 90% to 4% B, 70 to 90 min 4% B.

## Hydrogen/deuterium exchange analysis

Acidified deuterated methanol ($CD_3OD$, methanol-d4, Sigma-Aldrich; St. Louis, MO) was made by adding 1 μL of deuterated FA into 1 mL of $CD_3OD$. A total of 2 μL of the bioactive fractions were added into 18 μL of acidified methanol above; 15 μL of the mixture were analyzed by direct infusion into a modified 11 Tesla Fourier-transform mass spectrometer (FTMS; Thermo Scientific; Waltham, MA) using a NanoMate robot (Advion; Ithaca, NY) [37]. Full spectra were acquired with resolution set at 100 k.

## NMR analysis

Purified bioactive materials were dissolved in 250 μL of $CD_3OD$ and transferred into a 5-mm Shigemi NMR tube with a glass magnetic plug with susceptibility matched to $CD_3OD$ on the bottom. All NMR data were collected at 40˚C on an Agilent VNMRS 750 MHz spectrometer equipped with a 5 mm Varian indirect-detection probe with z gradient capability. Collected NMR data included 1 H spectrum, gradient selected correlation spectroscopy (gCOSY), TOCSY, NOESY with a mixing time of 500 ms, $^1H$-$^{13}C$ HSQC spectroscopy, and $^1H$-$^{13}C$ HMBC spectroscopy. The NMR spectra were analyzed using Mnova NMR software (Mestrelab Research, Spain).

## Determination of SPF concentration

The proton quantification experiments were performed at 23˚C on an Agilent 750 MHz VNMRS NMR spectrometer equipped with a 5 mm triple-resonance ($^1H$/$^{13}C$/$^{15}N$) indirect-detection probe with XYZ PFG gradient capability. The probe was calibrated using the qEstimate tool in the Agilent VnmrJ4.2 software with a known standard. The proton spectrum of the sample was collected with a 90˚ pulse angle of 8.5 ms, 16 scans, and 10.4 s delay between scans. The Agilent VnmrJ4.2 software was used to determine the concentration of the sample based on the integration values of proton peaks. A total of 5 well-resolved proton peaks (7.12 ppm [1 H], approximately 6.89 ppm [2 H], 4.41 [1 H], 3.83 [3H], and approximately 3.58 [2H]) was used, and the concentration of the sample was $1.55 \pm 0.07$ mM. All concentrations used in the cercarial paralysis assay were calculated based on this value.

## Cercarial paralysis assay

To capture the whole field while avoiding excess reflected light in a well, we used the lid of a 96-well plate (Costar; Corning, NY). A total of 40 μL of APW containing approximately 50 cercariae were added to each shallow well on the lid; 10 μL of SPF (dissolved in APW) was then added to reach the final concentration indicated. Using a high-speed camera (Olympus i-SPEED TR) attached to a stereomicroscope (Leica MZ125), we recorded cercariae movement at 20 to 60 fps at 1.25× magnification just prior to addition of test compounds until 3 to 4 min after treatment started. Raw movies were converted to.avi files using i-SPEED Viewer and compressed into JPEG format using ImageJ (addition of compound is considered time 0). We then counted the numbers of free-swimming or paralyzed cercariae every 5 s for 1 min and every 30 s thereafter for 3 min. The number of dead cercariae (those that never swim before and after SPF treatment) were subtracted from data. Experiments were performed in biological triplicate.

## Mouse infectivity assay

Swiss Webster mice (female) were purchased from Taconic Biosciences (Rensselaer, NY) and bred by RARC SPF Mouse Breeding Core (University of Wisconsin-Madison, Madison, WI). Mouse infections were performed by exposing mouse tails to *S. mansoni* cercariae according to standard protocol from BRI [24] with slight modifications. Briefly, we secured mice in rodent restrainers (Thomas Scientific, Cat #551-BSRR, Swedesboro, NJ) and put them vertically on top of a rack with grids. We pipetted 100 μL of each drug at proper concentration into a skinny glass tube (Fisher Scientific, Cat #14-958A, Hampton, NH) inside a 12 × 75 mm holding glass tube (VWR, Cat # 47729–570, Radnor, PA). A total of 300 μL of APW containing approximately 200 cercariae were pipetted into each skinny tube and incubated for 10 min before we inserted the mouse tail. Mouse tails were wiped with APW-moistened Kimwipes, inserted into the skinny tube, and exposed to cercariae for 30 min. The mouse tail was touching the bottom of the test tube unless otherwise specified. Six weeks post infection we euthanized these mice using pentobarbitol and perfused them according to standard protocols [24]. For each drug, we initially used 3 mice for controls (APW only) and 3 mice for each concentration tested except for 25 nM Ht-13-A and Ht-13-A-pr. We then repeated the experiments again with 3 mice for each condition. In addition to that, we included 6 mice for 25 nM Ht-13-A and Ht-13-A-pr.

Adult worms were recovered by hepatic portal vein perfusion, and males and females were unpaired by a brief incubation in 2.5% Tricaine (Sigma-Aldrich; St. Louis, MO) to facilitate counting. We counted total numbers of adult worms under a stereomicroscope (Leica MZ75). Livers from infected mice were fixed in 4% formaldehyde in PBS overnight. The largest liver lobes (left lobes) were submitted to the University of Wisconsin-Madison Histology Core Facility for sectioning and Hematoxylin and Eosin staining. Each left lobe was evenly cut into 4 to 6 pieces and paraffin embedded on a large cassette. One slide (4–6 liver sections) for each liver was used for histological examination, which provided a representative view throughout the whole liver lobe. We took a tiled image of the whole slide using a Zeiss Axio Zoom microscope and used ImageJ to determine the area of each section. Total numbers of eggs in each section were counted and normalized to the area.

In adherence to the Animal Welfare Act and the Public Health Service Policy on Humane Care and Use of Laboratory Animals, all experiments with and care of mice were performed in accordance with protocols approved by the Institutional Animal Care and Use Committee (IACUC) of the University of Wisconsin-Madison (protocol approval number M005569).

## Statistical analysis

GraphPad Prism (version 7) was used for all statistical analyses. One-way ANOVA test followed by Dunnett's multiple comparison test was used. Mean ± SD is shown in all figures.

## Supporting information

**S1 Fig. $^{1}$H NMR spectrum of SPF.** Peak areas of the nonoverlapping peaks were integrated and protons (δH 1.34, 3.10, 3.52, 3.56, 3.81, 4.40, 6.86, 6.90, and 7.09) showed integer ratios, supporting the mass spectrometry results that their signals were from the same compound. After adding the integration of overlapping peaks (δH 2.70, 2.72, 2.77, 2.79), a total of 19 protons were discovered, consistent with the best-fitting formula from the mass spectrometry results: ($C_{16}H_{20}N_2O_2$). NMR, nuclear magnetic resonance; SPF, Schistosome Paralysis Factor. (TIF)

**S2 Fig. FTMS determined the accurate *m/z* of the target molecule and revealed its isotopic pattern.** Before deuterium exchange (top panel), 273.1597 was the measured *m/z* of the target molecule. After deuterium exchange (bottom panel), *m/z* of the base peak increased to 275.1722 (deuterium singly charged target molecule with one proton replaced by deuterium), suggesting the presence of one exchangeable proton in SPF. FTMS, Fourier-transform mass spectrometry; SPF, Schistosome Paralysis Factor.
(TIF)

**S3 Fig. $^1$H-$^{13}$C HSQC spectroscopy NMR spectrum of SPF.** HSQC revealed the cross-correlation between directly bonded proton and carbon nuclei and determined the number of methyl, methylene, and methine groups. A total of 19 protons were attached to 11 carbons, including 3 methyl groups ($\delta$C 14.0, $\delta$H 1.34; $\delta$C 41.7, $\delta$H 2.71; $\delta$C 62.2, $\delta$H 3.81), 2 methylene groups ($\delta$C 29.0, $\delta$H 2.79, 3.56; $\delta$C 65.8, $\delta$H 2.70, 3.52), and 6 methine groups ($\delta$C 116.8, $\delta$H 6.86; $\delta$C 106.6, $\delta$H 6.90; $\delta$C 124.9, $\delta$H 7.09; $\delta$C 76.7, $\delta$H 3.10; $\delta$C 88.2, $\delta$H 4.40; $\delta$C 37.2, $\delta$H 2.77). The other 5 carbons that did not show up in the HSQC spectrum are the quaternary carbons. Based on the carbon chemical shift, the 2 methyl groups ($\delta$C 41.7, $\delta$H 2.71, and $\delta$C 62.2, $\delta$H 3.81) are likely to be bound to nitrogen and oxygen, respectively. HSQC, heteronuclear single quantum coherence spectroscopy; NMR, nuclear magnetic resonance; SPF, Schistosome Paralysis Factor.
(TIF)

**S4 Fig. TOCSY NMR spectrum of SPF.** TOCSY revealed that the aliphatic protons except the 2 methyl groups ($\delta$H 2.71 and 3.81) found binding to N and O in HSQC (S3 Fig) are from a single spin system. Cross-peaks were also observed among the aromatic proton $\delta$H 7.09 and the aliphatic protons ($\delta$H 3.56, 2.79 and 3.10) due to long-range couplings. HSQC, heteronuclear single quantum coherence spectroscopy; NMR, nuclear magnetic resonance; SPF, Schistosome Paralysis Factor; TOCSY, total correlation spectrometry.
(TIF)

**S5 Fig. COSY NMR spectrum of SPF.** Based on HSQC, protons 11 and 11′ ($\delta$H 2.79 and 3.56) are on the same carbon. Both have cross-peaks with proton 12 ($\delta$H 3.10) on COSY, which has an additional cross-peak with proton 13 ($\delta$H 4.40). This suggests $CH_2$ (C11, H11, and 11′)-CH (C12, H12)-CH (C13, H13) connectivity. Similarly, proton 14 ($\delta$H 2.77) is connected to CH (C13, H13). Methyl group $CH_3$ (proton 17, $\delta$H 1.34) and $CH_2$ group (proton 15, 15′, $\delta$H 2.70, 3.52) are directly connected to CH (proton 14). COSY, correlation spectroscopy; HSQC, heteronuclear single quantum coherence spectroscopy; NMR, nuclear magnetic resonance; SPF, Schistosome Paralysis Factor.
(TIF)

**S6 Fig. H-$^{13}$C HMBC NMR spectrum of SPF.** (A) Aliphatic region. Given the results from the COSY (S5 Fig) and the chemical shifts of C12 and C15 ($\delta$C 76.7 and 65.8), C12 and C15 are joined to a heteroatom. Because proton 18 ($\delta$H 2.71) has cross-peaks with both C12 and C15, it is a nitrogen atom that connects methyl group ($\delta$C 41.7, $\delta$H 2.71 on position 18), CH group ($\delta$C 76.7, $\delta$H 3.10), and $CH_2$ group ($\delta$C 65.8, $\delta$H 2.70 and 3.52). C13 has a chemical shift of 88.2 ppm, suggesting its connection to an oxygen. With HMBC, TOCSY, HSQC, and COSY, the connectivity of the aliphatic portions is resolved. (B) Aromatic region. The connectivity-built aliphatic structure has the formula $C_7H_{13}NO$, which leaves $C_9H_6NO$ after subtracting from the best-fitting formula. HSQC (S3 Fig) showed the existence of a methoxyl group ($\delta$C 62.2, $\delta$H 3.81). Therefore, the aromatic region was composed of $C_8H_3N$. HMBC data showed that 3 aromatic protons were located in different rings, implying a fused aromatic ring structure with one nitrogen. A substituted indole was the most common structure utilized in

organisms with the matching formula. In addition, HMBC showed that protons on the methoxyl group (δH 3.81) and the aromatic proton (δH 6.90) have cross-peaks with carbon (δC 143.1), suggesting they are meta to each other. The other proton (δH 6.86) was vicinal to proton (δH 6.90) because of their coupling seen in the COSY spectrum (S5 Fig). The aromatic singlet proton δH 7.09 showed cross-peaks with 3 aromatic carbons, 2 of those carbons (δC 120.6 and δC 138.1) had cross-peaks with protons (δH 6.86 and δH 6.90), respectively, consistent with an indole configuration. HMBC further confirmed C (δC 110.6) was linked to $CH_2$ (δH 2.79 and 3.56), and C (δC 143.7) was linked to the CH (δC 88.2, δH 4.40) across an oxygen atom. COSY, correlation spectroscopy; HMBC, heteronuclear multiple-bond correlation; HSQC, heteronuclear single quantum coherence spectroscopy; NMR, nuclear magnetic resonance; SPF, Schistosome Paralysis Factor; TOCSY, total correlation spectrometry.
(TIF)

**S7 Fig. NOESY NMR spectrum of SPF.** (A) Aliphatic region. The intensities of selected cross-peaks were integrated using Mnova software and shown in the spectrum. (B) Aromatic region. Results of the NOESY experiment support the final structures (Fig 2G and 2H) due to the presence of a NOE signal between H (δH 3.81) and H (δH 1.34), which could only be observed between protons with short spatial distance. For protons on the 3 consecutive chiral centers, H (δH 4.40) had an intense cross-peak with H (δH 2.77), whereas a weak signal was observed between H (δH 4.40) and H (δH 3.10) and no signal was observed between H (δH 2.77) and H (δH 3.10). This suggests that H (δH 4.40) and H (δH 2.77) are close to each other and both are distant from H (δH 3.10), which corresponds to (R, S, S) or (S, R, R) configuration on C 12, 13, 14 (δC 76.7, 88.2 and 37.2). This was further supported by NOESY signals between H (δH 2.79, 3.56) and the 3 H on chiral centers. H (δH 4.40) had a cross-peak with H (δH 2.79) but no cross-peak with H (δH 3.56). However, the opposite was observed for H (δH 3.10), which had a cross-peak with H (δH 3.56) but no cross-peak with H (δH 2.79). NMR, nuclear magnetic resonance; NOE, nuclear Overhauser effect; NOESY, nuclear Overhauser effect spectroscopy; SPF, Schistosome Paralysis Factor.
(TIF)

**S8 Fig. Stringent mouse infection experiment.** Numbers of adult worms recovered from mice exposed to approximately 100 cercariae that were pretreated with APW (N = 8), 2.5 μM SPF (N = 7), 2.5 μM Ht-13-A (N = 7), or 2.5 μM Ht-13-A-pr (N = 7). The mouse tail was lifted slightly during exposure so that its tip was 1 to 2 cm from the bottom of the test tube, avoiding direct contact with paralyzed cercariae. Data are mean ± SD, See S2 Data for corresponding raw data. APW, artificial pond water; SPF, Schistosome Paralysis Factor.
(TIF)

**S1 Table. Summary of protons and carbons from [1]H, COSY, HSQC, HMBC, and NOESY.** COSY, correlation spectroscopy; HMBC, heteronuclear multiple-bond correlation; HSQC, heteronuclear single quantum coherence spectroscopy; NOESY, nuclear Overhauser effect spectroscopy
(PDF)

**S1 Data. Raw data for cercarial paralysis assays (Fig 3).**
(XLSX)

**S2 Data. Raw data for mouse infectivity assays (Fig 4 and S8 Fig).**
(XLSX)

## Acknowledgments

*B. glabrata* snails were provided by the NIAID Schistosomiasis Resource Center of the BRI (Rockville, MD) through NIH-NIAID Contract HHSN272201700014I for distribution through BEI Resources. We thank Melanie Issigonis, Umair Khan, Jayhun Lee, and Tania Rozario for helpful discussions and comments on the manuscript; Tracy Chong, Jayhun Lee, and Janmesh Patel for help maintaining the schistosome life cycle; Melanie Issigonis for solving the pond water crisis; Björn Söderberg and Yanxing Jia for providing Ht-13-A, -B, and derivatives; Lingyang Zhu for expert assistance with NMR; Brian Imai for assistance with SPF purification; as well as Peg Stirewalt, James Leef, Tom Nerad, and Paul Mazzocchi for their early efforts to help solve this puzzle.

## Author Contributions

**Conceptualization:** Jiarong Gao, Fred A. Lewis, James J. Collins, III, Jonathan V. Sweedler, Phillip A. Newmark.

**Funding acquisition:** Jonathan V. Sweedler, Phillip A. Newmark.

**Investigation:** Jiarong Gao, Ning Yang, Fred A. Lewis, James J. Collins, III, Phillip A. Newmark.

**Methodology:** Jiarong Gao, Ning Yang, Fred A. Lewis, Peter Yau, James J. Collins, III, Jonathan V. Sweedler.

**Project administration:** Jonathan V. Sweedler, Phillip A. Newmark.

**Supervision:** Jonathan V. Sweedler, Phillip A. Newmark.

**Writing – original draft:** Jiarong Gao, Ning Yang, Jonathan V. Sweedler, Phillip A. Newmark.

**Writing – review & editing:** Jiarong Gao, Ning Yang, Fred A. Lewis, Peter Yau, James J. Collins, III, Jonathan V. Sweedler, Phillip A. Newmark.

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
