## [Editor Report · Decision Letter 0]

4 Jul 2019

Dear Phil, 

Thank you for submitting your manuscript entitled "A rotifer-derived paralytic compound prevents transmission of schistosomiasis to a mammalian host" for consideration as a Short Reports by PLOS Biology.

Your manuscript has now been evaluated by the PLOS Biology editorial staff, as well as by an academic editor with relevant expertise, and I'm writing to let you know that we would like to send your submission out for external peer review.

**Important**: Please also see below for further information regarding completing the MDAR reporting checklist. The checklist can be accessed here: https://plos.io/MDARChecklist

Please re-submit your manuscript and the checklist, within two working days, i.e. by Jul 08 2019 11:59PM.

Best wishes,

Roli

Senior Editor

PLOS Biology

INFORMATION REGARDING THE REPORTING CHECKLIST:

PLOS Biology is pleased to support the "minimum reporting standards in the life sciences" initiative (https://osf.io/preprints/metaarxiv/9sm4x/). This effort brings together a number of leading journals and reproducibility experts to develop minimum expectations for reporting information about Materials (including data and code), Design, Analysis and Reporting (MDAR) in published papers. We believe broad alignment on these standards will be to the benefit of authors, reviewers, journals and the wider research community and will help drive better practise in publishing reproducible research. 

We are therefore participating in a community pilot involving a small number of life science journals to test the MDAR checklist. The checklist is intended to help authors, reviewers and editors adopt and implement the minimum reporting framework. 

IMPORTANT: We have chosen your manuscript to participate in this trial. The relevant documents can be located here:

MDAR reporting checklist (to be filled in by you): https://plos.io/MDARChecklist

**We strongly encourage you to complete the MDAR reporting checklist and return it to us with your full submission, as described above. We would also be very grateful if you could complete this author survey:

https://forms.gle/seEgCrDtM6GLKFGQA

Additional background information:

Interpreting the MDAR Framework: https://plos.io/MDARFramework

Please note that your completed checklist and survey will be shared with the minimum reporting standards working group. However, the working group will not be provided with access to the manuscript or any other confidential information including author identities, manuscript titles or abstracts. Feedback from this process will be used to consider next steps, which might include revisions to the content of the checklist. Data and materials from this initial trial will be publicly shared in September 2019. Data will only be provided in aggregate form and will not be parsed by individual article or by journal, so as to respect the confidentiality of responses. 

Please treat the checklist and elaboration as confidential as public release is planned for September 2019.

We would be grateful for any feedback you may have.

---

## [Decision Letter · Decision Letter 1]

1 Aug 2019

Dear Dr Newmark,

I am writing on behalf of my colleague Senior Editor, Roland Roberts, who is the Editor Handling your manuscript as he is out on vacation this week.

Thank you very much for submitting your manuscript "A rotifer-derived paralytic compound prevents transmission of schistosomiasis to a mammalian host" for consideration as a Short Reports at PLOS Biology. Your manuscript has been evaluated by the PLOS Biology editors, an Academic Editor with relevant expertise, and by four independent reviewers. One of the reviewers chose to reveal his identity and his name appears with his review below. Please note that the fourth reviewer provided only brief comments as we wanted to move to a decision as expediently as was possible.

In light of the reviews (below), we are pleased to offer you the opportunity to address the comments from the reviewers in a revised version that we anticipate should not take you very long. Indeed, most of the comments can be addressed textually. However, the Academic Editor noted that addressing the comments from Reviewer #4 should not be too difficult by providing 16S amplification to check for the existence of a bacterial symbiont in the cultures, which we agree would strengthen your manuscript. As this is a Short Report, we do need you to keep the number of figures at no more than 4. Once you have revised, we will then assess your revision and your response to the reviewers' comments, and we may then consult the reviewers again.

Your revisions should address the specific points made by each reviewer. Please submit a file detailing your responses to the editorial requests and a point-by-point response to all of the reviewers' comments that indicates the changes you have made to the manuscript. In addition to a clean copy of the manuscript, please upload a 'track-changes' version of your manuscript that specifies the edits made. This should be uploaded as a "Related" file type. You should also cite any additional relevant literature that has been published since the original submission and mention any additional citations in your response. 

Before you revise your manuscript, please review the following PLOS policy and formatting requirements checklist PDF: http://journals.plos.org/plosbiology/s/file?id=9411/plos-biology-formatting-checklist.pdf. It is helpful if you format your revision according to our requirements - should your paper subsequently be accepted, this will save time at the acceptance stage.

Please note that as a condition of publication PLOS' data policy (http://journals.plos.org/plosbiology/s/data-availability) requires that you make available all data used to draw the conclusions arrived at in your manuscript. If you have not already done so, you must include any data used in your manuscript either in appropriate repositories, within the body of the manuscript, or as supporting information (N.B. this includes any numerical values that were used to generate graphs, histograms etc.). For an example see here: http://www.plosbiology.org/article/info%3Adoi%2F10.1371%2Fjournal.pbio.1001908#s5.

For manuscripts submitted on or after 1st July 2019, we require the original, uncropped and minimally adjusted images supporting all blot and gel results reported in an article's figures or Supporting Information files. We will require these files before a manuscript can be accepted so please prepare them now, if you have not already uploaded them. Please carefully read our guidelines for how to prepare and upload this data: https://journals.plos.org/plosbiology/s/figures#loc-blot-and-gel-reporting-requirements.

Upon resubmission, the editors assess your revision and assuming the editors and Academic Editor feel that the revised manuscript remains appropriate for the journal, we may send the manuscript for re-review. We aim to consult the same Academic Editor and reviewers for revised manuscripts but may consult others if needed.

We expect to receive your revised manuscript within one month. Please email us (plosbiology@plos.org) to discuss this if you have any questions or concerns, or would like to request an extension. At this stage, your manuscript remains formally under active consideration at our journal; please notify us by email if you do not wish to submit a revision and instead wish to pursue publication elsewhere, so that we may end consideration of the manuscript at PLOS Biology.

When you are ready to submit a revised version of your manuscript, please go to https://www.editorialmanager.com/pbiology/ and log in as an Author. Click the link labelled 'Submissions Needing Revision' where you will find your submission record. 

Sincerely,

Emma Ganley PhD

Chief Editor, PLOS Biology

eganley@plos.org

On behalf of:

Senior Editor

PLOS Biology

Reviewer remarks:

-----------------

Reviewer #1

-----------------

This paper takes an observation made almost 40 years ago that rotifers produce a substance that interferes with schistosome cercariae ability to infect experimental animals. The authors identify the reactive component from the rotifer Rotaria rotatoria as Schistosome Paralysis Factor, demonstrate that it paralyzes the cercariae, and can prevent cercariae from infecting mice. They also perform a number of biophysical experiments to work out a potential structure of the molecule as a tetracyclic alkaloid.

The research is well done and the paper well written.

A few minor points:

1. Lines 109-111; Please clarify. An epimer differs at one carbon but R,S suggest enantiomers. Are you just trying to say that ht-13-A and thus maybe SPF are racemic? Alternatively are you trying to say that there are R & S forms and one is responsible for the paralysis.

2. Lines 114-115; Please clarify. 30 min exposure to their tails? Did you show or just suggest that SPF works on the tails of cercariae? Did you count tails that had dislodged from cercariae? Were the heads moving on paralyzed cercariae?

3. Line 147; produced by the rotifer

-----------------

Reviewer #2

-----------------

Review of manuscript PBIOLOGY-D-19-01881R1 entitled “A rotifer-derived paralytic compound prevents transmission of schistosomiasis to a mammalian host” by Gao, Sweedler, Newmark and others. 

This is a very well-written paper that builds upon Stirewalt and Lewis’s 1981 finding that association of intermediate host snails with rotifers impeded output, motility and infectivity of schistosome cercariae emerging from them. This study has many significant strengths, including, identification of Rotaria rotatoria as the source rotifer species for the biological activity, logical and definitive schema for isolating and characterizing a tricyclic alkaloid that the authors dubbed Schistosome Paralysis Factor (SPF), careful structure activity analyses using SPF and two related alkaloids naturally occurring in Streptomyces bacteria along with structural analogs of these, and a careful analysis of dose-dependent decrements in infectivity in cercariae exposed to SPF and the Streptomyces alkaloids. The latter analysis supports that exposure to SPF in a fresh water environment could block the ability of schistosome cercariae to infect hosts and bolsters the authors conclusion that SPF could form the basis of an environmental or other treatment to prevent transmission of schistosome infection. At least two significant and testable hypotheses are framed in this paper to stimulate future studies. One is that the microbiomes of rotifers associated with intermediate host snails, rather than the rotifers themselves could be the source of the paralytic alkaloids. This hypothesis is plausible given the paralytic activities of structurally related alkaloids of prokaryotic origin. Similarly, the authors provide a strong basis for a hypothesis that SPF acts on serotonin signaling pathways necessary for normal motility in schistosome cercariae. 

I noted only one minor substantive organizational shortcoming in the paper that related to the appropriate point to highlight the presence of a serotonin backbone in SPF and the bacterial alkaloids. Also, there were only a few glitches in copy editing and clarity in this very well-written manuscript.

SUBSTANTIVE ISSUES

The observation that SPF contains a serotonin backbone comes rather “out of the blue” in the Conclusion. For this reviewer, this is one of the most important findings in the paper, being consistent with previous reports of the importance of serotonin signaling in schistosome motility and providing a jumping off point for investigations of the mode of action of SPF against schistosome cercariae. Given this, shouldn’t this point be introduced in Results and Discussion? At or around line 90, where the possible structures of SPF are introduced would seem to be a logical point for this introduction. It might also be helpful to shade the serotonin backbone in the structures of SPF, ht-13-A, ht-13-A pr and ht-13-B in Fig. 3. 

MINOR ISSUES OF COPY EDITING AND CLARITY

Line 147: Insert “by” between “produced” and “the rotifer”.

Line 162: Insert “a” before “100 um cell strainer”. 

Line 205: Suggest substituting “using a” for “through”. 

Lines 254-255: I assume the Tricaine incubation was to unpair the adult worms for counting, correct? If so, I suggest “…recovered by hepatic portal vein perfusion, and males and females were unpaired by a brief incubation in 2.5% Tricaine to facilitate counting.”

-----------------

Reviewer #3: Jon Clardy

-----------------

This in an interesting manuscript with findings of interest to fields as varied as chemical ecology and infectious disease. In brief, the authors have characterized a compound produced by a rotifer that paralyzes the snail stage of the schistosome responsible for schistosomiasis, a neglected disease of growing importance. The general outline of the story: the ability of some but not all rotifers to produce a schistosome paralysis factor (SPF) that render the cercariae life-cycle stage of the parasite ‘paralyzed’ – unable to infect human hosts. 

The manuscript fleshes out this bare bones picture in significant ways. Most importantly it uses bioassay-guided fractionation to establish that SPF is a small molecule and uses chemical analysis – primarily mass spectrometry (MS) and nuclear magnetic resonance (NMR) to characterize SPF. All of the work described has been done at an acceptable technical level, and the procedures are described so that they could be readily reproduced. The findings reported in this manuscript form a solid basis for pursuing further studies both on small molecule signaling and therapeutic agents. I recommend publication with no changes.

-----------------

Reviewer #4

-----------------

I thought the paper was interesting, though the biosynthetic origin of the tetracyclic alkaloid (called Schistosome Paralysis Factor, SPF) should really be discussed in the manuscript. Other than SPF, the Streptomyces-derived alkaloids ht-13-A and ht-13-B are the only two examples of naturally occurring 3,4-oxepino-fused indoles, and thus it seems likely that the SPF is also bacterial origin, rather than “produced by the rotifer Rotaria rotatoria”, as the abstract suggests.

Some experiments to explore a potential bacterial origin of SPF would be in order. Did the authors check for the presence of any Streptomyces spp in Rotaria rotatoria? Sequencing would provide a quick means to get an overview of associated microbiota.

---

## [Editor Report · Decision Letter 2]

29 Aug 2019

Dear Phil,

Thank you for submitting your revised Short Report entitled "A rotifer-derived paralytic compound prevents transmission of schistosomiasis to a mammalian host" for publication in PLOS Biology. The Academic Editor and I have now assessed your revisions and responses to reviewers, and we're delighted to let you know that we're now editorially satisfied with your manuscript.

However before we can formally accept your paper and consider it "in press", we also need to ensure that your article conforms to our guidelines. A member of our team will be in touch shortly with a set of requests. As we can't proceed until these requirements are met, your swift response will help prevent delays to publication.

IMPORTANT: Please could you also address my Data Policy requests below, namely, provide underlying numerical data for some of the Figure panels, and cite the location of the data in respective legends?

Please note that you may have the opportunity to make the peer review history publicly available. The record will include editor decision letters (with reviews) and your responses to reviewer comments. If eligible, we will contact you to opt in or out.

Early Version: Please note that an uncorrected proof of your manuscript will be published online ahead of the final version, unless you opted out when submitting your manuscript. If, for any reason, you do not want an earlier version of your manuscript published online, uncheck the box. Should you, your institution's press office or the journal office choose to press release your paper, you will automatically be opted out of early publication. We ask that you notify us as soon as possible if you or your institution is planning to press release the article.

Best wishes,

Roli

Senior Editor

PLOS Biology

ETHICS STATEMENT:

The Ethics Statements in the submission form and Methods section of your manuscript should match verbatim. Please ensure that any changes are made to both versions.

-- Please include the full name of the IACUC/ethics committee that reviewed and approved the animal care and use protocol/permit/project license. Please also include an approval number if one was obtained.

-- Please include the specific national or international regulations/guidelines to which your animal care and use protocol adhered. Please note that institutional or accreditation organization guidelines (such as AAALAC) do not meet this requirement.

-- Please include information about the form of consent (written/oral) given for research involving human participants. All research involving human participants must have been approved by the authors' Institutional Review Board (IRB) or an equivalent committee, and all clinical investigation must have been conducted according to the principles expressed in the Declaration of Helsinki.

DATA POLICY:

Regardless of the method selected, please ensure that you provide the individual numerical values that underlie the summary data displayed in Figs 3ABCDEFG, 4BCDEFG and S8, as they are essential for readers to assess your analysis and to reproduce it. Please also ensure that figure legends in your manuscript include information on where the underlying data can be found.

---

## [Editor Report · Decision Letter 3]

13 Sep 2019

Dear Dr Newmark,

On behalf of my colleagues and the Academic Editor, Chaitan Khosla, I am pleased to inform you that we will be delighted to publish your Short Reports in PLOS Biology. 

PRESS 

Kind regards,

Alice Musson

Publication Assistant, 

PLOS Biology

on behalf of

Roland Roberts,

Senior Editor

PLOS Biology